# The Effects of Copolymer Compatibilizers on the Phase Structure Evolution in Polymer Blends—A Review

**DOI:** 10.3390/ma14247786

**Published:** 2021-12-16

**Authors:** Ivan Fortelný, Josef Jůza

**Affiliations:** Institute of Macromolecular Chemistry of the Czech Academy of Sciences, Heyrovského náměstí 2, CZ 162 06 Praha 6, Czech Republic; fortelny@imc.cas.cz

**Keywords:** compatibilization efficiency, copolymers, interfacial tension, droplet breakup, coalescence, blend morphology

## Abstract

This paper summarizes the results of studies describing the effect of block and graft copolymers on the phase structure formation and evolution in immiscible polymer blends. The main phenomenological rules for prediction of the copolymer compatibilization efficiency are briefly described and compared with selected experimental data. The results of the theories of equilibrium distribution of a copolymer between the blend interface and the bulk phases and its effect on the blend interfacial tension are summarized. The theories of the compatibilizer effect on the droplet breakup in flow are analyzed. The mechanisms of the copolymer effect on the coalescence of droplets in flow are compared and their effect on the droplet size is shown. The problems of reliable description of the effect of a copolymer on the coalescence in quiescent state are presented. Obstacles to derivation of a realistic theory of the copolymer effect on the competition between the droplet breakup and coalescence are discussed. Selected experimental data are compared with the theoretical results.

## 1. Introduction

Many polymer blends are incompatible. This means that their components show large interfacial tension, which leads to rough phase structure and poor mechanical properties of these blends. Therefore, they need to be made compatible for their successful applications in practice [1,2,3]. Well known and generally accepted knowledge is that block and graft copolymers with blocks identical to, miscible with or adhering to the related components of a polymer blend can serve as compatibilizers [1,2,3,4]. A basic principle of the effect of copolymers on the morphology of polymer blends, which is tendency of copolymers to be localized at the interface with the blocks oriented to the related homopolymers, is also well understood. However, a much more peculiar task is the proper choice of the structure of the copolymers for achieving the optimum structure and properties of a given blend. In spite of a huge number of the theoretical and experimental studies on the effect of the molecular structure of added copolymer to the morphology and properties of the polymer blends, no general rule for prediction of the compatibilization efficiency of copolymers has been formulated so far. It seems that an analysis of the effect of a copolymer on all individual steps of the phase structure formation of polymer blends is necessary for full elucidation of this problem.

The correct description of the phase structure evolution in the flowing polymer blends containing a compatibilizer is extremely difficult. Mutual connection between the area of the interface and interfacial tension in the compatibilized polymer blends is an additional problem. This problem has to be solved in addition to those joined with describing the phase structure evolution in immiscible binary polymer blends [5,6,7,8,9]. It should be mentioned that, from the thermodynamic point of view, flowing immiscible polymer blends are in a transient or steady state but not in the equilibrium. Therefore, using of the rule of the minimum Gibbs energy for the prediction of any aspect of the blend morphology evolution, e.g., the distribution of a copolymer between the bulk phases and the interface, is not rigorous. The principle of the minimum entropy production was formulated for the steady state [10], but it has not been formulated adequately for the flowing polymer blends so far. Therefore, the effects of a compatibilizer on the individual steps of the phase structure evolution have been studied. Most of studies in this field are focused on blends with droplets in the matrix morphology, where the description of the phase structure evolution in the binary blends is better developed than for blends with the co-continuous or other morphologies. Therefore, this paper mostly deals with the effect of a copolymer on the phase structure evolution in the blends with the droplets in the matrix morphology. It is assumed that the morphology of compatibilized blends, formed in the melt, is more-or-less frozen during cooling at common processing procedures. Therefore, this review summarizes the results of the studies of the effect of copolymers on the morphology of polymer blends in melt independently of the structure (crystalline or amorphous) of their components in the solid state.

This paper is focused on the analysis of results of the theoretical studies of: (i) a copolymer distribution between the bulk phases and the interface in compatibilized blends, (ii) the effect of a copolymer on the droplet breakup, (iii) the effect of a copolymer on the flow-induced coalescence, (iv) the competition between the droplet breakup and coalescence in steady flow of compatibilized blends and (v) the effect of a copolymer on the coalescence in quiescent compatibilized blends. The fundamental conclusions following from the results of the experimental studies are confronted with the available theoretical results. The main aim of this review is to evaluate the available theories and the experimental data with respect to the prediction of the effect of the copolymer molecular structure on the fineness and stability of the morphology of molten compatibilized blends. This paper does not discuss a reactive compatibilization, where the block or graft copolymers are formed during flow. The course of the chemical reaction should be taken into account in addition to the problems mentioned above in describing the reactive compatibilization.

## 2. Phenomenological Predictive Rules

Some papers proposed predictive rules based on experimental data without an analysis of individual effects of a copolymer on the phase structure evolution in compatibilized polymer blends. Lyngaae-Jørgensen [11] formulated the theory for the effect of shear stress on the stability of the droplet size in the polymer blends containing a compatibilizer. This theory is based on the addition of the elastic entropy to the Gibbs energy of mixing. The author’s rheo-optical measurements of the stability of the droplet dimensions in the flowing compatibilized polystyrene/polymethylmethacrylate blends corresponded with the theory. However, this theory does not predict the effect of copolymer amount and architecture on the droplet size in compatibilized blends.

Tang and Huang [12] assumed that the change in the interfacial tension of a compatibilized blend, *γ*, with the concentration of a compatibilizer, *c*, is proportional to *γ*_0_–*γ*_*s*_, where *γ*_*s*_ is the interfacial tension in a blend with the interface fully saturated with the copolymer, while *γ*_0_ is the interfacial tension between both homopolymer phases without a compatibilizer. The following equation was derived [12] using the above assumption:(1)$$\gamma =(\gamma_{0}-\gamma_{s})e^{-Kc}+\gamma_{s},$$
where *K* is the parameter of proportionality and *e* is the Euler number. The authors assumed that the capillary numbers in compatibilized and non-compatibilized blends were equal to the same function of the viscosity ratio of the dispersed phase and matrix. Therefore, the effect of a copolymer on the size of dispersed droplets is controlled by its effect on the interfacial tension. This approach disregards flow-induced coalescence, although it fundamentally affects the difference between the droplet size in blends with and without a compatibilizer for blends with the finite concentration of a dispersed phase.

Kim et al. [13] proposed that the effect of the block copolymer on the droplet size was controlled by the ratio, *s*_r_, of the swelling powers of the block copolymer segments at the interface outside the droplet versus those inside the droplet. The swelling power *s_ij_* of the homopolymer *i* in the copolymer block *j* is defined as *s_ij_* = *N_j_*/*P_i_* − 2 *χ**_ij_**N_j_*, where *N_j_* and *P_i_* are the number of segments in the block *j* and homopolymer *i*, respectively, and *χ**_ij_* is the Flory–Huggins interaction parameter between the block *j* and the homopolymer *i*. The results for the poly(cyclohexyl methacrylate)/styrene-acrylonitrile copolymer blends compatibilized with the poly(styrene-b-methyl methacrylate) block copolymer showed that (i) the internal emulsification failure (micelles formed inside the droplets) appeared for *s*_r_ < 0.4, (ii) the external emulsification failure (micelles formed in the matrix) manifested for *s*_r_ > 2.5, (iii) the unstable emulsification (droplets coalesce slowly at low shear rates) was detected for 1 > *s*_r_ > 0.4 and (iv) the stable emulsification (coalescence at low shear rates was not found) appeared for 2.5 > *s*_r_ > 1.

Chun and Han [14] proposed that the compatibilization efficiency of blocks copolymers was controlled by their order–disorder transition temperature *T*_ODT_ (temperature of the transition between microphase-separated and uniform molecular structure of copolymers). They stated that the block copolymers behaved as efficient compatibilizers for the immiscible polymer blends only if the blending temperature was above their *T*_ODT_. This statement follows from their experimental results for the polystyrene/polyisoprene/styrene-butadiene block copolymer and the polypropylene/polystyrene/styrene–hydrogenated–butadiene–styrene block copolymer blends.

## 3. Distribution of a Copolymer in Polymer Blends and the Interfacial Tension

### 3.1. Calculation of the Equilibrium Distribution of a Copolymer between the Interface and Bulk Phases

The first question to be answered before the calculation of the effect of a copolymer on the breakup and coalescence of droplets is which part of the added copolymer is localized at the interface. It is a self-consistent problem because the amount of a copolymer controls the interfacial tension which substantially affects the interfacial area in the blend. The first step in the solution of this problem should be the determination of the copolymer fraction localized at the interface in the blends with a certain interfacial area in the volume unit. As mentioned in the Introduction, no rule relating to the minimum Gibbs energy for an equilibrium has been formulated for the polymer blends in a steady or even transient state. All theories predicting the distribution of a copolymer between the interface and bulk phases and the copolymer effect on the interfacial tension in the immiscible polymer blends have been derived for the thermodynamic equilibrium. Fortunately, the available experimental data for many polymer blends indicate that the equilibrium copolymer distribution can serve as a reasonable approximation for the blends in steady flow [13,15,16,17,18,19].

Many theoretical studies have been devoted to the prediction of the amount of a copolymer at the interface and the interfacial tension in immiscible polymer blends in the thermodynamic equilibrium, e.g., [20,21,22,23,24,25,26,27,28,29,30,31,32,33,34,35,36,37]. Some of these studies were based on Leibler’s model of the interface [20]. Leibler assumed a brush of the A–B block copolymer at the interface in the blend of strongly immiscible A, B homopolymers. He used the Flory–Huggins theory for calculation of an equilibrium between the chemical potential of a copolymer localized at the interface and in the bulk phase. This model was applied to the blends containing copolymers with various architectures and the *χ* parameters between the copolymer blocks and the relating homopolymers [21,23,24,25,26,29,38]. Noolandi [22] pointed out that an enthalpic contribution of the copolymer molecules to the interfacial tension reduction was omitted in the original Leibler’s theory [20]. Based on this consideration, a new term was added to the expression for Gibbs energy of a copolymer at the interface [21,39].

A more general self-consistent field theory based on the expression of the free energy of a system as functional was proposed by Noolandi with collaborators [40]. This theory was also applied to various equilibrium systems with a reasonable agreement with the experiments [27,28,30,31,34,35,36,37]. It was found [30] that an increase in the copolymer chemical potential was accompanied by an increase in the concentration of copolymer chain at the interface and by a decrease in the interfacial tension. The free energy associated with the interface between a segregated polymer layer and a homopolymer matrix can give rise to an attractive interaction between the segregated layers. Semenov [41] proposed an analytical mean-field theory of the copolymer segregation at the interface of the immiscible polymers. Noolandi’s and Semenov’s theories consider a more realistic model of compatibilized polymer blends than the Leibler theory. On the other hand, Leibler’s theory provides an easier insight to the effect of the architecture of a copolymer and the *χ* interaction parameters between the homopolymers and copolymer blocks on the copolymer distribution and the interfacial tension in the blend.

The concentration of a copolymer at the interface and the interfacial tension in most of the papers [20,21,23] mentioned above was calculated assuming that the amount of the copolymer at the interface did not affect this concentration in bulk phases. This assumption is justified if the volume of the interfacial layer in a system is negligible with respect to the volumes of the bulk phases, e.g., when interpreting the measurement of the interfacial tension by the pendant drop method [21,23]. However, this assumption is not generally applicable to compatibilized polymer blends showing fine phase structure, such as the blends containing a moderate concentration of submicron particles. The assumption that all copolymer molecules were localized at the interface was made in some other papers [24,25]. This assumption also cannot be applied to polymer blends generally.

### 3.2. Theories Considering the Effect of Finite Fraction of a Copolymer at the Interface on Its Distribution in Blends

Fortelný and Jůza [39] derived the theory of the distribution of the A–B block copolymer between the interface and bulk phases in immiscible A/B blends and of the effect of the block copolymer on the blend interfacial tension. They used Leibler’s model and its Noolandi’s modification with consideration of the effect of the amount of a copolymer on the interface on copolymer concentration in bulk phases. They derived transcendent equations for concentration of a block copolymer at the interface for a dry brush [20] (the copolymer blocks short with respect to the homopolymer chains are stretched at the interface), a wet brush [20] (the relatively long copolymer blocks are stretched at the interface) and a wet mushroom [21] (the mutually non-interacting blocks of a copolymer rarely occupying the interface). Unfortunately, these equations have no analytical solution and they have to be solved numerically. The copolymer concentrations at the interface obtained using the above procedure can be substituted to the explicit equations for the change of the interfacial tension. The conditions for these approximations are compared in Table 1.

Later, this theory was generalized to strongly immiscible A/B blends compatibilized with C-D block copolymers where the C and D blocks are miscible with or adhering to the related A and B homopolymers [42]. It was assumed that the absolute values of the interaction parameters *χ*_AC_ and *χ*_BD_ were much smaller than *χ*_AB_. Using of the original Leibler’s approach for calculation of the volume fraction of a copolymer at the interface, *ϕ*_CD_^(I)^, leads to the following equations for the wet brush [42]
(2)$$\begin{array}{l} 5\left({\left(\frac{3}{32}\right)}^{\frac{1}{3}}\left(\frac{N_{C}}{P_{A}^{2/3}}+\frac{N_{D}}{P_{B}^{2/3}}\right)-{\left(\frac{2}{9}\right)}^{\frac{1}{3}}\left({\left(P_{A}\right)}^{\frac{1}{3}}\chi_{\mathrm{AC}}N_{C}+{\left(P_{B}\right)}^{\frac{1}{3}}\chi_{\mathrm{BD}}N_{D}\right)\right)\Psi^{\frac{2}{3}}+ln\left(N_{C}\Psi \right)+ln\left(N_{D}\Psi \right)=\;\\ \ln\frac{\left({\tilde{\varphi }}_{\mathrm{CD}}-{\tilde{\varphi }}_{\mathrm{CD}}^{\left(I\right)}\right)}{\left(1+{\tilde{\varphi }}_{B}\exp\left\{-\left(\chi_{\mathrm{BD}}-\chi_{\mathrm{AD}}\right)N_{D}-\left(\chi_{\mathrm{BC}}-\chi_{\mathrm{AC}}\right)N_{C}\right\}\right)}+\left(\chi_{\mathrm{AD}}-\chi_{\mathrm{BD}}\right)N_{D}+\left(\chi_{\mathrm{AC}}-\chi_{\mathrm{AC}}\right)N_{C}+\frac{N_{C}}{P_{A}}+\frac{N_{D}}{P_{B}}-2 \end{array},$$
dry brush [42]
(3)$$\frac{9}{2}N\Psi^{2}+ln\left(N_{C}\Psi \right)+ln\left(N_{D}\Psi \right)=ln\frac{\left({\tilde{\varphi }}_{\mathrm{CD}}-{\tilde{\varphi }}_{\mathrm{CD}}^{\left(I\right)}\right)}{\left(1+{\tilde{\varphi }}_{B}\exp\left\{-\left(\chi_{\mathrm{BD}}-\chi_{\mathrm{AD}}\right)N_{D}-\left(\chi_{\mathrm{BC}}-\chi_{\mathrm{AC}}\right)N_{C}\right\}\right)}+\chi_{\mathrm{AD}}N_{D}+\chi_{\mathrm{AC}}N_{C}-2,$$
and wet mushroom [42] (a small concentration of a copolymer at the interface) models
(4)NC3/2PA+ND3/2PBΨ+lnNCΨ+lnNDΨ−NCPA+NDPB=lnφ˜CD−φ˜CDI1+φ˜Bexp−χBD−χADND−χBC−χACNC+χAD−χBDND−5,
where *N*_C_ and *N*_D_ are numbers of segments with length *a* in blocks C and D, *N* = *N*_C_ + *N*_D_, *P*_A_ and *P*_B_ are numbers of segments with length *a* in homopolymers A and B, *χ**_ij_* is *χ* parameter between the homopolymer *i* and the block *j*, *ϕ*_CD_ is the volume fraction of a copolymer in the blend. A swung dash above symbols for volume fractions generally indicates their ratio to the volume fraction of homopolymer A, *ϕ*_A_, eg. ${\tilde{\varphi }}_{\mathrm{CD}}$ = *ϕ*_CD_/*ϕ*_A_, and the parameters *Γ* and *Ψ* are defined as [39,42]
*Γ*=*aS*/*ϕ*_*A*_(5)
and [42]
(6)$$\Psi =\frac{{\tilde{\varphi }}_{\mathrm{CD}}^{\left(I\right)}}{N\Gamma }=\frac{\varphi_{\mathrm{CD}}^{\left(I\right)}}{NaS},$$
where *S* is the interfacial area per volume of a blend.

The *Γ* parameter reflects the surface to volume ratio. Although it does not occur in the equations in this article due to the substitution of Equation (6) into original equations, it was one of the input parameters in the papers cited [39,42].

The following transcendent equations (with results shown in Figure 1) were derived using Noolandi’s modification of the Leibler approach for the wet brush [42]
(7)$$\begin{array}{l} 5\left({\left(\frac{9}{32}\right)}^{\frac{1}{3}}\left(\frac{N_{C}}{P_{A}^{2/3}}+\frac{N_{D}}{P_{B}^{2/3}}\right)-{\left(\frac{2}{9}\right)}^{\frac{1}{3}}\left({\left(P_{A}\right)}^{\frac{1}{3}}\chi_{\mathrm{AC}}N_{C}+{\left(P_{B}\right)}^{\frac{1}{3}}\chi_{\mathrm{BD}}N_{D}\right)\right)\Psi^{\frac{2}{3}}+ln\left(N_{C}\Psi \right)+ln\left(N_{D}\Psi \right)+ln\left[\frac{2\cdot \chi_{\mathrm{AB}}^{1/2}}{\pi }6^{1/6}\left(\frac{N_{C}}{P_{A}^{1/3}}+\frac{N_{D}}{P_{B}^{1/3}}\right)\Psi^{1/3}\right]=\;\\ \ln\frac{\left({\tilde{\varphi }}_{\mathrm{CD}}-{\tilde{\varphi }}_{\mathrm{CD}}^{\left(I\right)}\right)}{\left(1+{\tilde{\varphi }}_{B}\exp\left\{-\left(\chi_{\mathrm{BD}}-\chi_{\mathrm{AD}}\right)N_{D}-\left(\chi_{\mathrm{BC}}-\chi_{\mathrm{AC}}\right)N_{C}\right\}\right)}+\left(\chi_{\mathrm{AD}}-\chi_{\mathrm{BD}}\right)N_{D}+\frac{N_{C}}{P_{A}}+\frac{N_{D}}{P_{B}}-\frac{7}{3} \end{array},$$
dry brush [42]
(8)$$\begin{array}{l} \frac{9}{2}N\Psi^{2}+ln\left(N_{C}\Psi \right)+ln\left(N_{D}\Psi \right)+ln\left(\frac{{\left(24\chi_{\mathrm{AB}}^{}\right)}^{\frac{1}{2}}}{\pi }N\Psi \right)\;\\ =ln\frac{\left({\tilde{\varphi }}_{\mathrm{CD}}-{\tilde{\varphi }}_{\mathrm{CD}}^{\left(I\right)}\right)}{\left(1+{\tilde{\varphi }}_{B}\exp\left\{-\left(\chi_{\mathrm{BD}}-\chi_{\mathrm{AD}}\right)N_{D}-\left(\chi_{\mathrm{BC}}-\chi_{\mathrm{AC}}\right)N_{C}\right\}\right)}+\chi_{\mathrm{AD}}N_{D}+\chi_{\mathrm{AC}}N_{C}-3 \end{array},$$
and wet mushroom [42] models
(9)$$\begin{array}{l} \left(\frac{N_{C}^{3/2}}{P_{A}}+\frac{N_{D}^{3/2}}{P_{B}}\right)\Psi +ln\left(N_{C}\Psi \right)+ln\left(N_{D}\Psi \right)+ln\left[\frac{{\left(24\chi_{\mathrm{AB}}^{}\right)}^{1/2}}{\pi }\left(N_{C}^{\frac{1}{2}}+N_{D}^{\frac{1}{2}}\right)\right]=\;\\ =\ln\frac{\left({\tilde{\varphi }}_{\mathrm{CD}}-{\tilde{\varphi }}_{\mathrm{CD}}^{\left(I\right)}\right)}{\left(1+{\tilde{\varphi }}_{B}\exp\left\{-\left(\chi_{\mathrm{BD}}-\chi_{\mathrm{AD}}\right)N_{D}-\left(\chi_{\mathrm{BC}}-\chi_{\mathrm{AC}}\right)N_{C}\right\}\right)}+\left(\frac{N_{C}}{P_{A}}+\frac{N_{D}}{P_{B}}\right)+\left(\chi_{\mathrm{AD}}-\chi_{\mathrm{BD}}\right)N_{D}-5 \end{array}.$$

The dependence of the copolymer content at the interface ${\tilde{\varphi }}_{\mathrm{CD}}^{\left(I\right)}$ given by Equations (7)–(9) on the total copolymer content ${\tilde{\varphi }}_{\mathrm{CD}}$ is demonstrated in Figure 1.

The solutions of the transcendent Equations (2)–(4) can be substituted into the following explicit equations for the difference between the interfacial tensions of a compatibilized and a neat blend, *γ* −*γ*_0_, derived using Leibler’s approach for the wet brush [42]
(10)$$\frac{\gamma -\gamma_{0}}{kT}=-\frac{1}{a^{2}}\left\{\left[{\left(\frac{3}{4}\right)}^{\frac{1}{3}}\left(\frac{N_{C}}{P_{A}^{2/3}}+\frac{N_{D}}{P_{B}^{2/3}}\right)-{\left(\frac{4}{3}\right)}^{\frac{2}{3}}\left(P_{A}^{\frac{1}{3}}\chi_{\mathrm{AC}}N_{C}+P_{B}^{\frac{1}{3}}\chi_{\mathrm{BD}}N_{D}\right)\right]\Psi^{\frac{5}{3}}+2\Psi \right\},$$
dry brush [42]
(11)$$\frac{\gamma -\gamma_{0}}{kT}=-\frac{1}{a^{2}}\Psi \left\{2+3N\Psi^{2}\right\},$$
and wet mushroom models [42]
(12)$$\frac{\gamma -\gamma_{0}}{kT}=-\frac{\Psi }{a^{2}}\left\{2+\frac{1}{2}\left(\frac{N_{C}^{3/2}}{P_{A}}+\frac{N_{D}^{3/2}}{P_{B}}\right)\Psi \right\},$$
where *k* is the Boltzmann constant and *T* is the absolute temperature.

The following equations were derived [42] using Noolandi’s modification of the Leibler theory for the wet brush
(13)$$\gamma -\gamma_{0}=-\frac{kT}{a^{2}}\left\{\frac{7}{3}\Psi +\left[\frac{3^{1/3}}{2^{2/3}}\left(\frac{N_{C}}{P_{A}^{2/3}}+\frac{N_{D}}{P_{B}^{2/3}}\right)-\frac{2^{4/3}}{3^{2/3}}\left(P_{A}{}^{\frac{1}{3}}\chi_{\mathrm{AC}}N_{C}+P_{B}{}^{\frac{1}{3}}\chi_{\mathrm{BD}}N_{D}\right)\right]\Psi^{\frac{5}{3}}\right\}$$
and dry brush models [42]
(14)$$\gamma -\gamma_{0}=-\frac{kT}{a^{2}}\left\{3\Psi +3N\Psi^{3}\right\}.$$

Equation (12) with ${\tilde{\varphi }}_{\mathrm{CD}}^{\left(I\right)}$ given by Equation (9) is valid for the wet mushroom model.

The dependence of the relative decrease in the interfacial tension *γ* given by Equations (12)–(14) on the total copolymer content ${\tilde{\varphi }}_{\mathrm{CD}}$ is shown in Figure 2.

The numerical solution of the above equations showed that the differences between results of related equations of the original Leibler’s theory and its Noolandi’s modification varied from negligible to remarkable, depending on the system parameters [39,42]. However, no qualitative differences between these results were found. For a certain polymer blend, an adequate model for describing a copolymer distribution and of its effect on the interfacial tension should change with increasing $\varphi_{\mathrm{CD}}^{\left(I\right)}$ from the wet mushroom (small concentration of a copolymer) to the wet brush or dry brush in dependence on *N_i_*/*P_i_* ratio. Since equations for the wet mushroom, wet brush, and dry brush are approximate, being derived for the different models of the interfacial layer, the transition between them need not be continuous. The numerical calculations showed that the fraction of the added copolymer localized at the interface changed over a broad range depending on the system parameters. Therefore, this fraction must be calculated for individual systems for reliable prediction of the interfacial tension in the compatibilized polymer blends.

It was found [39] that an increasing length of the homopolymers or the copolymer blocks led to an increase in the fraction of the copolymer localized at the interface. The copolymers with the block length somewhat larger than the chains of the homopolymers are most efficient in reduction of the interfacial tension. For the block substantially longer than the homopolymers, the reduction of the interfacial tension starts to decrease with an increasing ratio of the lengths of the blocks and the related homopolymer chains. A compatibilization efficiency (characterized by the interfacial tension) depends on the combination of the *χ* parameters between the blocks and the related homopolymers [42]. The dependence of a decrease in the interfacial tension on the block length of symmetric A-B copolymers is shown in Figure 3.

For the block C miscible with the homopolymer A and the block D identical to the homopolymer B, the largest reduction in the interfacial tension was found for *N*_C_ slightly smaller than *N*_D_. For the block C miscible with the homopolymer A and the block D slightly immiscible with the homopolymer B, the maximum decrease in the interfacial tension is for *N*_C_ larger than *N*_D_. However, a decrease in the interfacial tension is smaller than this for A ≡ C and B ≡ D.

A problem for the above theory is the existence of copolymer concentration regions where the conditions for application neither the wet mushroom nor the dry or wet brush are met. Therefore, a new procedure based on calculations of the added amount of a copolymer from the amount of the copolymer at the interface and its brush thickness was proposed [43]. This theory enables avoiding of the approximations used in calculating the Gibbs energy and the thickness of the brush for the individual models. The brush thickness equal to the square root of the quadratic end-to-end distance for the Gaussian chain is assumed for non-stretched chains at the interface. Thickness of the brush of stretched chains was given by minimum Gibbs energy of the brush in the Leibler theory [20] with the remaining parameters constant. It was found that this method of calculation substantially limited discontinuities between concentrations of a copolymer at the interface and between interfacial tensions calculated for models of small and large amounts of a copolymer at the interface. Note that in many cases, a small amount of a copolymer (the interface is occupied by non-stretched copolymer molecules) already causes a decrease in the interfacial tension to zero.

### 3.3. The Effect of a Copolymer Geometry on the Interfacial Tension in Compatibilized Polymer Blends

A decrease in the interfacial tension was calculated for the addition of copolymers with various geometry. Lyatskaya et al. [26] studied the effect of A–B block and random copolymers on the interfacial tension of A/B blends using self-consistent field (SCF) theory. Their theory was based on the lattice model and Monte Carlo computer simulation assuming that the volume fraction of a copolymer in bulk phases is constant. They found that the diblock copolymers reduced the interfacial tension more efficiently than the random copolymers of the same length. The effect of a random copolymer on the blend interfacial tension is limited by phase separation of the copolymer as its own phase. Efficiency of the random copolymers decreases with growing difference of their composition from 50:50. For the mixture of 50% of 60:40 and 50% of 40:60 random copolymers, larger reduction in interfacial tension was found than for 50:50 random copolymer. Alternating copolymers have the same effect on the interfacial tension as 50:50 random copolymers of the same length.

The Leibler theory [20] and the SCF theory were applied to the calculation of the effect of graft and star copolymers [26]. For the same total number of segments, diblock copolymers are more efficient than graft and star copolymers. Efficiency of graft copolymers decreases with the increasing number of teeth. Lyatskaya and Balazs [29] studied the effect of mixtures of: (a) diblock copolymers with different molecular weights, (b) diblock and comb copolymers and two comb copolymers (the former with a B-backbone and A-teeth, the latter with an A-backbone and B-teeth). They found that a mixture of two symmetric diblock copolymers was more efficient in the reduction in the interfacial tension than one diblock with an average molecular weight. Mixtures of block and comb copolymers reduced the interfacial tension more efficiently than the neat block or the comb copolymers. Additionally, a mixture of the comb copolymers with changed comb and teeth was more efficient than its individual components.

The effect of a number of teeth in comb copolymers on the reduction in interfacial tension was studied by Lyatskaya et al. using the theory based on Leibler’s model [28]. It was found that the graft copolymers with a small number of long teeth were more efficient compatibilizers than those with a large number of short teeth. Similar conclusions also followed from Monte Carlo computer simulations by Gersappe et al. [33]. Calculations described in this paragraph disregard the reduction in the copolymer concentration in the bulk phases caused by a localization of a part of copolymer chains at the interface. This point is important for a substantial part of the polymer blends with the fine phase structure. Noolandi showed [44] that the multiblock copolymers could be more efficient in the reduction of the interfacial tension than a diblock or triblock if their length of block was sufficient for their substantial extension into the related homopolymer phase.

### 3.4. The Effect of a Copolymer Micellization on the Interfacial Tension in Compatibilized Polymer Blends

The above equations were derived assuming the equilibrium between the block copolymer chains dissolved in the bulk phases and them localized in the copolymer brush at the interface. Therefore, the theory is applicable only to the blends with the non-negligible solubility of copolymer chains at least in one of bulk phases. When this assumption is not valid, small submicron pieces of a copolymer in one or both bulk phases should appear. The theory [39,42,43] does not consider a formation of micelles if the concentration of a copolymer in one of the bulk phases achieves the critical micelle concentration. For the calculation of the chemical potential of a copolymer in the bulk phases, which should be equal to the chemical potential of a copolymer in the micelles, the solution of the relevant equation of Equations (2)–(4) or (7)–(9) should be used. It is generally impossible to determine the critical micelle concentration directly from the amount of added copolymer.

The micelle formation in the compatibilized polymer blends and in homopolymer hosts was studied quite extensively (e.g., [21,23,36,41,45,46]), but with neglecting a decrease in the concentration of a copolymer in the bulk phases due to coverage of the interface by copolymer, which generally should not have been omitted for polymer blends. It is assumed that three forms of micelles can be formed in dependence on the ratio of A block to the whole copolymer lengths, *f*_A_ = *N*_A_/*N*. It is expected [31] that spherical micelles are formed for *f*_A_ < 0.13, cylindrical for 0.13 < *f*_A_ < 0.35 and lamellar for *f*_A_ > 0.35. The following explicit equations were proposed for the chemical potentials of spherical, $\mu_{\mathrm{Mic}}^{\mathrm{sph}}$, cylindrical, $\mu_{\mathrm{Mic}}^{\mathrm{cyl}}$, and lamellar, $\mu_{\mathrm{Mic}}^{\mathrm{lam}}$, micelles assuming that their corona was not swollen with homopolymer chains [23].
(15)$$\frac{\mu_{\mathrm{Mic}}^{\mathrm{sph}}}{kT}={\left(\frac{3}{2}\right)}^{\frac{4}{3}}f_{A}^{\frac{4}{9}}{\left(1.74f_{A}^{-\frac{1}{3}}-1\right)}^{\frac{1}{3}}{\left(\chi N\right)}^{\frac{1}{3}}$$
(16)$$\frac{\mu_{\mathrm{Mic}}^{\mathrm{cyl}}}{kT}=1.19{\left(\chi f_{A}N\right)}^{\frac{1}{3}}{\left(1.64-\ln f_{A}\right)}^{\frac{1}{3}}$$
(17)$$\frac{\mu_{\mathrm{Mic}}^{\mathrm{lam}}}{kT}=0.669{\left(\chi N\right)}^{1/3}{\left(5.64-f_{A}\right)}^{1/3}$$

This assumption is plausible if the homopolymer chain is longer than the copolymer block forming the corona. The value *f*_A_ = 0.29 instead of 0.35 was proposed by Semenov [41] for starting formation of the lamellar micelles. From the kinetic considerations, the author assumed that micelles started to form at concentrations of the copolymer dissolved in the bulk phases larger than those resulting from equations describing thermodynamic equilibrium. Based on the same reasons, he assumed that spherical micelles were formed initially even if cylindrical or lamellar micelles should be formed in the thermodynamic equilibrium. The dependence of all three chemical potentials given by Equations (15)–(17) on the *f*_A_ is demonstrated in Figure 4.

Unfortunately, no explicit equation for the chemical potentials of the micelles having the corona swollen with the homopolymer is available. Whitmore and Noolandi [47] derived a theory of formation of the spherical micelles of the block copolymer in the homopolymer that contains the chains identical to one block considering swelling of the corona (the block A of the copolymer AB forms a core and the block B forms a corona when the copolymer is dissolved in the polymer B). A precise calculation of the critical micellar concentration was performed only numerically. They [47] proposed the following approximate equation for the critical volume fraction of the dissolved copolymer, $\varphi_{\mathrm{AB}}^{\left(\mathrm{cr}\right)}$
(18)$$\varphi_{\mathrm{AB}}^{\left(\mathrm{cr}\right)}≅0.30\frac{\chi N}{{\left(\chi N_{A}\right)}^{2/3}}\exp\left\{-\chi N_{A}+2.475{\left(\chi N_{A}\right)}^{1/3}+\frac{0.78}{{\left(\chi N_{A}\right)}^{1/6}}-\frac{3}{2}+\frac{1}{2}\left[\alpha_{B}^{2}+\frac{2}{\alpha_{B}}-3\right]\right\}$$
where the last square bracket in Equation (18) expresses the contribution of stretching of the B blocks of a copolymer in the micelle corona. *α*_B_ is defined as
(19)$$\alpha_{B}={\left(\frac{3}{N_{B}}\right)}^{1/2}\frac{l_{B}}{a}$$
where *l*_B_ is the corona thickness. For small *N*_B_, *α*_B_ ≅ 1; unfortunately for large *N*_B_, no analytical expression for *α*_B_ is available. It follows from numerical calculations that $\varphi_{\mathrm{AB}}^{\left(\mathrm{cr}\right)}$ increases with increasing *N*_B_ and decreasing *P*_B_.

Chang and Morse [36] calculated the critical micelle concentration and its effect on the interfacial tension in compatibilized polymer blends combining the SCF and Helfrich theories. They found that the zero interfacial tension could not be achieved in A/B blend compatibilized with A–B block copolymer in contrast to Leibler’s result [20]. This is due to the consideration of two facts. The first is the possibility of swelling of the B-core of the micelles with the homopolymer B dissolved in the A-rich phase. The second is the equality of the chemical potentials of all components in the B-rich core and in the B-rich macroscopic phase. Leibler et al. [48] proposed a simple microscopic model of spherical micelle formation considering the equilibrium between the dissolved copolymer chains and the finite number of micelles in the blend. Adedeji et al. [38] modified the above theory [48] for the blends compatibilized with the block copolymers containing the blocks miscible with related homopolymers; they found that the degree of exothermic mixing between the copolymer block and the homopolymer was important for the blend morphology.

Note that micelles start to form in the phase where concentration of a copolymer is larger, i.e., in the phase identical to longer block in A/B blends compatibilized with A–B block copolymer. Therefore, *f*_A_ ≤ 0.5 for the micelles of A-B copolymer starting to form in homopolymer B. For the A/B blends compatibilized with the C–D copolymer, the interaction parameters between the copolymer blocks and the related homopolymers besides the block lengths affect starting of micelle formation.

Retsos et al. [23] calculated the conditions for micelle formation in the blends compatibilized with the graft copolymers by the methods used for the derivation of Equations (14)–(16) considering the differences in the architecture between the block and studied graft copolymers.

## 4. The Effect of a Compatibilizer on the Droplet Breakup

### 4.1. Formulation of the Problem

It is well known that the droplet deformation and breakup is controlled by the capillary number, *Ca*, defined as a ratio of stress in flow to the interfacial stress (ratio of interfacial tension and the radius of the non-deformed droplet) [5,6,8,9]. For the shear flow
(20)$$Ca=\frac{\eta_{m}GR}{\gamma }$$
where *η*_m_ is the viscosity of the matrix, *G* is the shear rate and *R* is the droplet radius.

Therefore, the effect of a compatibilizer on the interfacial tension and, therefore, on *Ca*, must be considered. As it follows from the preceding section, the explicit expressions for the wet brush, dry brush and small amount of a copolymer at the interface were derived. However, these expressions contain the value of concentration of a copolymer at the interface, which is a solution of the equations solvable only numerically. Besides the change in the equilibrium interfacial tension, the effect of nonuniform distribution of a copolymer on the droplet surface caused by the flow must be considered [6,9,49,50]. The copolymer distribution on the droplet interface is controlled by the competition between the surface convective flux promoting the concentration gradient and the surface diffusion flux that tends to restore the homogeneous copolymer distribution along the interface (see Figure 5). Moreover, a diffusion of the copolymer chains from or to areas at the interface—where the interfacial tension is substantially lower or higher than the equilibrium interfacial tension—and the bulk phases is generally possible.

Droplet breakup can appear only for *Ca* larger than its critical value *Ca*_c_. For the Newtonian droplets with the clean interface in a Newtonian matrix, *Ca*_c_ is a function of only the viscosity ratio, *p*. The dependence on *p* of the critical droplet radius, related to *Ca*_c_ through Equation (20), was experimentally determined by Gabriele et al. [51]. Elasticity parameters are variables further affecting *Ca*_c_ in the blends containing viscoelastic components [5,6,7,8,9]. There are two main mechanisms of the droplet breakup: stepwise for *Ca* only slightly larger than *Ca*_c_ and transient for *Ca* >> *Ca*_c_. Repeated breakup into two halves is characteristic for the stepwise mechanism. For the transient mechanism, a drop is elongated into a long fiber, which bursts into a number of small droplets. The approach to the derivation of the equations describing the effect of a copolymer on droplet breakup is the same as for the more intensively studied effect of the low-molecular-weight surfactants on droplet breakup in Newtonian emulsions [52,53]. However, the results of these studies cannot be directly applied to the compatibilized polymer blends due to qualitative differences in the values of the rheological functions and in relations between the concentration of a compatibilizer at the interface and the interfacial tension [54].

If a Newtonian droplet covered with a copolymer having position-dependent density *q* in a Newtonian matrix is considered, the stress boundary condition on its surface is described by the following equation [6,49,53]
(21)$$n\cdot T_{m}-pn\cdot T_{d}=\frac{\gamma (q)}{\eta_{m}GR_{0}}n(\nabla_{s}\cdot n)-\frac{1}{\eta_{m}GR_{0}}\nabla_{s}\gamma (q)$$
where **T**_m_ and **T**_d_ are the stress tensors in the matrix and dispersed phase, respectively, **n** is the unit normal directed outward from the droplet, *p* is the viscosity ratio of the droplets and matrix, *γ*(*q*) is the actual interfacial tension, *R*_0_ is the radius of non-deformed spherical droplet, *G* is the shear rate, and ∇_s_ is the surface gradient operator ∇_s_ = (**I** − **nn**)⋅∇, where **I** is the unit tensor. If a transport of a copolymer between the interface and bulk phases can be neglected, the changes in *q* at the interface are described by the equation [6,49,52,55]:(22)$$\frac{\partial q}{\partial t}+\nabla_{s}\left(qu_{s}-\frac{1}{Ca_{eq}M}\nabla_{s}q\right)+q(\nabla_{s}\cdot n)(u\cdot n)=0$$
where
(23)$$M=\frac{\gamma_{\mathrm{eq}}R_{0}}{\eta_{m}D_{s}}$$
and *Ca*_eq_ is the capillary number for a droplet having the equilibrium interfacial tension *γ*_eq_, **u** is the flow velocity, **u**_s_ is the velocity components tangential to the interface [**u**_s_= (**I** − **nn**)⋅**u**], and *D*_s_ is the surface diffusivity of the copolymer.

The solution of Equation (21) for large droplet deformation needed for breakup is not easy; it cannot be obtained in an analytical form even for a system with a neat interface where the interfacial tension is constant [9]. Solving of Equations (21) and (22) is complicated by the coupling of the copolymer distribution with the droplet shape, which depends upon the detailed copolymer distribution. Moreover, as is discussed above, no explicit expressions for *q* as a function of the amount of the added copolymer and the interfacial area are available and the present knowledge of the copolymer surface diffusivity *D*_s_ is strongly limited.

### 4.2. Available Results

Yang et al. [56] experimentally studied the diffusion of the polystyrene-*b*-polyisoprene block copolymer at the interface of polyisoprene of various molecular weights with dimethylformamide (non-solvent for polystyrene). They found that the interfacial diffusion coefficient decreased very slowly with the molecular weight of the polyisoprene in agreement with the theoretical considerations and for the system under study it was of the order of tenth of µm^2^/s. The authors estimated that the ratio of the bulk and interfacial viscosities was of the order 10^2^. It has led us to the assumption that neglecting of migration of the copolymer chains between the interface and bulk phases due to local changes in *q* is a reasonable approximation.

Flumerfelt [52] generalized the Cox theory [9,57] of the deformation and orientation of a Newtonian droplet in a Newtonian matrix, considering the migration of a surfactant along the interface and between the interface and bulk phases. Stone and Leal [55] studied the finite deformation and breakup of the droplet in a system with the surfactant insoluble in both the droplet and matrix phases. The total amount of a surfactant at the interface is a constant, while *q* can change due to an increase in the interfacial area caused by the droplet deformation and due to the flow-induced inhomogeneity of the copolymer chains at the interface. Equations (21) and (22) were solved for the small difference between *γ* and *γ*_eq_ [55]. The explicit expressions for *q* and the deformation parameter of the droplet were derived for the extensional flow assuming *Ca*_eq_ << 1 and *p* << 1. However, these expressions were derived for the relation between *γ* and *q* typical of a low-molecular-weight surfactant. This relation strongly differs from those for the compatibilized polymer blends.

In the paper [49], the equations for *q* and the deformation parameter were recalculated using the equation *γ*_eq_ = *γ*_0_ − *fq*_eq_^3^, which relates to the dry brush system. For *fq*_eq_^3^/*γ*_0_ → 0 or *M* → 0, the results reduce to the Taylor equations for a system without a compatibilizer. There is no effect of a compatibilizer on the interfacial tension in the first case. In the second case, the gradient of the compatibilizer concentration is not induced because flow-induced changes in compatibilizer concentration are eliminated by diffusion. The derivation of the relating results for the relations between *γ*_eq_ and *q*_eq_ valid for the wet brush system and the system with a small amount of a copolymer at the interface is straightforward. However, no analytic expression for *q*_eq_ is available for any model of a copolymer brush. The above conditions for the transition of the results for compatibilized to those for the non-compatibilized blends (*γ*_eq_ → *γ*_0_, *M* → 0) are valid independently of the flow geometry and the relation between *γ*_eq_ and *q*_eq_.

Equations (21) and (22) were derived for the compatibilized system containing Newtonian components. To the best of our knowledge, no theory describing the droplet breakup in compatibilized blends containing viscoelastic components has been derived so far. The above discussion shows that a quantitative theoretical prediction of the effect of a compatibilizers on characteristics of the stepwise droplet breakup is impossible at the present state of the art. Therefore, the further discussion is based on a qualitative theoretical consideration confronted with the experimental results.

Two main parameters characterizing the stepwise breakup are the critical capillary number and the breakup frequency [9]. The further discussion is focused on *Ca*_c_, because understanding of the breakup frequency is limited even to the blends without a compatibilizer. It follows from the assumption that *Ca*_c_ for a compatibilized blend is the same as for a blend without a compatibilizer with the same rheological properties that critical droplet radius for breakup, *R*_c_, in the same flow field should scale as *γ*_eq_/*γ*_0_. This is not the case of polymer blends [8]. Generally, the Marangoni effect, caused by a gradient compatibilizer concentration at the interface induced by flow, and a dilution of a copolymer at the interface due to the droplet deformation, enhances the droplet resistance to the deformation and breakup. This consideration is supported by the experimental data [58,59,60,61,62] in contrast with small deformations of droplets scaling with *Ca*_eq_ [63]. Therefore, *Ca*_c_ for the compatibilized blends is larger than those for the related non-compatibilized blends. The presence of a compatibilizer can change the droplet breakup mechanism. Flow-induced accumulation of the compatibilizer molecules at the drop tips was observed [64], which can lead to the tip streaming of small droplets [58].

Due to an increase in *Ca*_c_, a decrease in the critical droplet radius, compared for the same flow fields, in a compatibilized blend with respect to the non-compatibilized blend with the same rheological parameters is smaller than that which relates to a decrease in interfacial tension [8,50,59].

A difference between the interfacial tension of a system of elongated fibers and of spherical droplets in the equilibrium has to be considered in the analysis of a transient breakup [49]. Palierne and Lequeux [65] generalized Tomotika’s theory [66] of the cylinder breakup to a viscoelastic cylinder covered by a compatibilizer in the viscoelastic matrix at rest. Generally, the breakup time, *t*_B_, of a cylinder increases with the decreasing interfacial tension. An analysis of Palierne and Lequeux’s theory [65] showed [49] that *t*_B_ for compatibilized blend increased somewhat more rapidly than it related to the same decrease in *γ* for a non-compatibilized blend. As a consequence, a larger number of smaller daughter droplets appear in flow after *t*_B_. It should be mentioned that the relations between *γ* and *q* valid for the low-molecular-weight surfactants are used in [65], which must be substituted by those for the copolymers. Therefore, the problems discussed above complicate also using of the Palierne and Lequeux theory [65] for a quantitative description of a transient breakup in the compatibilized polymer blends. The further complication is a possibility of the diffusion of a copolymer from the bulk phases to the thread surface during its elongation, which cannot be a priori neglected for large *t*_B_.

## 5. The Effect of a Compatibilizer on the Coalescence

The course of the coalescence of two dispersed droplets is commonly divided into four steps [9,67]:1.Approaching of the droplets;2.Drainage of the continuous phase trapped between the droplets, possibly deformed by the axial force;3.Rupture of the rest of continuous phase after the droplet approach to the critical distance, *h*_c_, usually by the formation of a “hole” on the thinnest spot;4.Evolution of the “neck” to form a coalesced droplet.

Note that splitting of the first two steps is generally artificial because the inter-droplet hydrodynamic interaction is a long-range one. However, the inter-droplet interaction at short distances seems to be decisive and the hydrodynamic interaction between the approaching droplets is not frequently considered.

To describe the approach of spherical droplets in the simple shear and extensional flows is easy when the inter-droplet hydrodynamic interaction is not considered. There is no difference between the droplets with a neat surface and droplets with the surface covered by a compatibilizer. Therefore, coalescence efficiency, i.e., the ratio of the rate of coalescence in a certain system to the rate of coalescence in a related one where the droplets do not interact till their touching, is considered. For the polymer blends without a compatibilizer, it is assumed that the rupture of the thin matrix film is very fast. The available theories do not consider the possibility that the thin “neck” linking the coalescing droplets bursts due to the axial force having the direction of their movement away during their further rotation. Therefore, most theories of the flow-induced coalescence in the blends without a compatibilizer are focused on the competition between the approach of the droplets to *h*_c_ and the rotation of the droplet doublet to the position where the direction of the axial inter-droplet forces causes droplets to start moving droplets away [6,9,68]. The presence of a compatibilizer on the droplets affects the drainage of the matrix trapped between them and also can affect the course of rupture of the rest of the continuous phase at the droplet approach to the critical distance.

Two main mechanisms of the coalescence suppression by a compatibilizer were proposed, as shown in Figure 6: a decrease in the mobility of the interface due to the Marangoni effect (mass transfer along an interface, in this case caused by the resistance to inhomogeneous compatibilizer distribution induced by matrix flow), slowing drainage of the matrix [50,69,70], and the effect of the steric repulsive force (induced by stretching of touching copolymer blocks) which oppose the approach of the droplets to the distance necessary for the matrix film rupture [71,72,73,74,75]. The matrix drainage between the colliding droplets in the compatibilized blends is affected by an increase in the size of the deformed droplet area with the decreasing interfacial tension [6,9,67]. Because the droplet deformation strongly affects the matrix drainage [9], a decrease in the interfacial tension caused by a compatibilizer contributes to a decrease in the coalescence efficiency. Generally, all three effects should be considered in evaluating the coalescence efficiency in the compatibilized polymer blends.

Milner and Xi [69] considered the Marangoni effect during the collision of spherical droplets covered with a block copolymer. They assumed that the copolymer was compressed at a half of the droplet surface and neglected the diffusion of the copolymer along the interface. The calculated repulsive force from the copolymer compression was incorporated into the theory of coalescence proposed by Wang et al. [76]. The Milner and Xi theory overestimates the value of the Marangoni stress because the area from which the copolymer is displaced is much smaller than half of the droplet surface for the inter-droplet distance substantially smaller than their radius. The finite rate of the copolymer diffusion also leads to a decrease in the Marangoni stress. On the other hand, this theory does not consider the droplet deformation which leads to a strong decrease in the coalescence efficiency.

Hudson et al. [77] analyzed the experimentally determined efficiency of the coalescence in compatibilized blends assuming the Marangoni effect and considering solubility of a surfactant. They compared importance of the diffusion-limited sorption with the surface diffusion. Their analysis showed that the results of the theories based on the assumption that the surfactant was insoluble (used in, e.g., [69]) differed qualitatively from the experimental observations. They concluded that the coalescence efficiency for small capillary numbers correlated with the product of the Marangoni number and the effective Peclet number, accounting for the diffusive and sorption processes. For the large capillary numbers, the coalescence efficiency correlated with the product of the effective Marangoni number, gauging the importance of the Marangoni effect during the thin film drainage, with the thin-film interfacial Peclet number.

A number of studies on various aspects of the Marangoni effect on the matrix drainage between the spherical droplets exist in the literature [78,79,80,81]. The similar aspect of the Marangoni effect was also studied for the highly deformed droplets where the diameter of the deformed part of the droplets was substantially larger than the inter-droplet distance [70]. In both cases, the Marangoni stress, leading to a decrease in the coalescence efficiency, increases with the shear rate *G*. This increase is slowed down by an increase in the diffusion coefficient of a copolymer along the interface. The limited knowledge of a copolymer diffusion mentioned above complicates quantitative prediction of the contribution of Marangoni effect to the coalescence. For constant *G*, the Marangoni effect manifests itself in the coalescence efficiency as an increase in the viscosity of the droplets.

A steric repulsion of the compatibilized droplets manifests itself at an inter-droplet distance relating to the thickness of the copolymer brushes on the droplet surfaces, which scales with the copolymer radius of gyration. It was shown that the minimum block copolymer coverage for full inhibition of coalescence was independent of the shear rate and reciprocally proportional to the molecular weight of the outward copolymer blocks [42,73]. The discussion of the contribution of the Marangoni effect and steric repulsion to the coalescence in flowing compatibilized polymer blends was provided in papers [6,8,82]. It should be mentioned that the Marangoni stress starts to affect the approach of the droplets at much longer distances (scaled with a droplet radius) than those where the steric repulsion can appear. The Marangoni effect leads to dilution of a copolymer on the top of approaching droplets. Therefore, a negative correlation should exist between the contribution of the Marangoni effect and that of the steric repulsion to the coalescence suppression. The quantitative contribution of the above mechanisms cannot be calculated due to limited knowledge of copolymer diffusion along the interface and non-existing theory of the steric repulsion when the coverage of the droplet surface with copolymer is lower than that necessary for the complete coalescence suppression. The effect of a compatibilizer on the critical inter-droplet distance *h*_c_ is important for the correct interpretation of the course of coalescence. However, this problem has not been adequately addressed so far. It can be assumed that a more important contribution to the coalescence suppression is the Marangoni effect for a low droplet coverage with a compatibilizer and the steric repulsion for a high droplet coverage. The Marangoni effect is compared with the steric repulsion in Table 2.

A further contribution to the coalescence suppression is an easier droplet deformation due to a decrease in the force needed to droplet deformation caused by a decrease in interfacial tension [82], because the coalescence efficiency of deformed droplet strongly decreases with respect to that for spherical droplets [9]. A reliable quantitative prediction of this effect is complicated by the fact that expression for the radius of deformed area was derived for a neat droplet surface. Generally, it can be concluded that the role of individual mechanisms in the coalescence suppression cannot be quantified at the present state of the art. All the above-discussed mechanisms contribute to the coalescence suppression. As it follows from the experimental data, the effect of a compatibilizer is substantially more pronounced in the coalescence suppression than in the reduction of the critical capillary number [8]. Therefore, the effect of a compatibilizer on the polymer morphology manifests itself more strongly in the polymer blends with a high content of the dispersed phase than in those containing a small amount of the dispersed phase.

So far, little attention has been paid to the theoretical analysis of the effect of a compatibilizer on the coalescence in a quiescent state. It was found that approaching of the droplets in the quiescent blends was induced by the inter-droplet molecular forces and the Brownian motion [83]. The effect of a compatibilizer on the droplet Brownian motion should be small, but the inter-droplet molecular forces can be affected substantially by a compatibilizer. However, no theoretical analysis of this effect is available. The effects of a compatibilizer on the drainage of the continuous phase between approaching droplets are discussed above. It should be considered that the approaching of the droplets is slow in quiescent blends due to weak attractive forces. Therefore, the approach of the undeformed droplets through the whole trajectory can be considered also in compatibilized systems. On the other hand, the resistance of the drained matrix against the droplets approach can slow down the coalescence but it cannot fully inhibit it due to central character of the attractive molecular forces.

## 6. The Competition between Breakup and Coalescence in Compatibilized Blends

When describing the competition between the droplet breakup and coalescence in compatibilized polymer blends, it should be considered that the breakup increases and the coalescence decreases the interfacial area available for a copolymer. The calculation of the dependence of the dispersed droplet size on its concentration in the compatibilized polymer blends needs knowledge of a relation between the interfacial tension and the whole amount of a copolymer in the system and of the effects of a copolymer on the droplet breakup and on the coalescence. As it follows from the above discussion, approximate theories of these effects are only available and they do not provide explicit expressions suitable for describing the competition between the droplet breakup and coalescence. Therefore, it is not surprising that attempts to calculate the droplet size in compatibilized blends are limited.

The steady droplet size in flowing compatibilized blends was calculated from the balance between the droplet breakup and coalescence in [84]. For a relation between the interfacial tension and the density of a copolymer at the interface, the approximate expression of Lyatskaya et al. [26] for the wet brush model was used. The approximate theory of the coalescence by Janssen and Meijer [85] with the coalescence time proposed by Jeelani and Hartland [70] was chosen to describe the coalescence probability. The frequency of the droplet breakup was described by an equation containing an unspecified function, *f*, of rheological properties of the blend components, introduced in [86]. It was assumed that the compatibilizer affected the droplet breakup only through the effect of the interfacial tension on the capillary number (constant *f* was assumed). The effect of a compatibilizer on the dependence of the average droplet radius on the droplet concentration was calculated for three types of systems: (1) with a constant density of a copolymer at the interface, (2) with all copolymer molecules localized at the interface and (3) with limited maximum copolymer density at the interface. Two factors limit the applicability of this theory to prediction of the droplet size in the compatibilized polymer blends: (a) the approximation used in describing the droplet breakup and coalescence (b) together with necessity to know the amount of a copolymer at the interface, having a complex relation to the amount of a copolymer added to the system (see above).

## 7. Discussion of Some Experimental Results with Respect to the Available Theories

This section is focused on the discussion of some experimental results originally interpreted without a deep theoretical analysis. We believe that, in spite of the impossibility of a quantitative prediction of the effect of a compatibilizer on the phase structure of polymer blends, a confrontation of the obtained experimental data with the available theoretical results can contribute to better understanding of the compatibilizer effects on the phase structure of polymer blends. On the other hand, an enormous number of experimental studies of the compatibilizer effect has been made so far. Moreover, many of their results were obtained for systems which did not reflect simplifications used in theoretically treated models, e.g., the same length of segments of all blend components, simple shear or extensional flow, the effect of cooling (crystallization), [16,17,18,19,87]. Therefore, only a small part of experimental results, which can be plausibly compared with available theories, are discussed here.

### 7.1. Emulgation Curves

Favis and his group proposed emulsification curves describing the effect of the amount of a compatibilizer on the size of the dispersed particles in polymer blends [5,88,89,90,91]. They compared dependences of the average size of the dispersed particles on the compatibilizer concentration related to the amount of the dispersed phase for various blend compositions. The minimum average droplet diameter, *d*_min_, was achieved at a certain copolymer concentration, *C*_crit_, after which *d* was independent of the amount of the added copolymer. Three basic types of emulsification curves [91,92] (see Figure 7) were proposed, which were interpreted as follows: When both the *d*_min_ and *C*_crit_ are independent of the concentration of the dispersed phase, all copolymer molecules are localized at the interface and the flow-induced coalescence is totally suppressed at *C*_crit_ (Type 1); When *d*_min_ increases and *C*_crit_ decreases with the concentration of the dispersed phase, whole copolymer migrates to the interface, but the copolymer does not suppress the flow-induced coalescence completely (Type 2); When *d*_min_ and *C*_crit_ increase with the concentration of the dispersed phase, not whole copolymer migrates to interface due to the micelle formation and the coalescence is not fully suppressed (Type 3).

It follows from Leibler’s theory of the equilibrium copolymer distribution between the interface and bulk phases [20,39,42] that the volume fraction of a copolymer at the interface and a decrease in the interfacial tension depend on the volume fraction of the dispersed phase only through the droplet size (parameter *Γ*) (see Section 3, Equations (7)–(14)) if formation of micelles is not considered. If it is assumed that this theory is adequate for polymer blends in a steady flow, the type of emulsification curves is not affected by the dissolution of a part of the added copolymer in the bulk phases and the real interfacial area occupied by the copolymer molecule cannot be determined from *d*_min_. Similarly, the start of micellization does not depend on the volume fraction of the dispersed phase in a blend. On the other hand, the rate of decrease in a number of the copolymer molecules migrating to the interface depends on the volume fraction of the bulk phase, where the micelles start to form. The emulsification curves generally conformed to the assumption that the adsorbed compatibilizer molecules reduce the interfacial tension, causing a decrease in the size of the dispersed droplets, till the interface saturation. After that, further addition of a copolymer does not affect the fineness of the phase structure.

### 7.2. The Effect of a Copolymer Molecular Weight on Its Compatibilization Efficiency

Adedeji et al. [15] studied the poly(cyclohexyl methacrylate)/poly(methyl methacrylate) (PCHMA/PMMA) (80/20) blends compatibilized with poly(styrene-*b*-methyl methacrylate) (PS-*b*-PMMA). They found that PS-*b*-PMMA copolymer started to form micelles in bulk phases before saturation of the blend interface. This result does not contradict the assumption that the equilibrium distribution of a copolymer in the blends can be applied to the description of the compatibilized blends at steady mixing. Generally, the start of the micelles formation is controlled by the critical micelle concentration of a copolymer in the bulk phase which can be achieved before the interface saturation (see Section 3).

The effect of a compatibilizer on the size of dispersed droplets and its stability during annealing was studied by Macosko et al. [72] in the polystyrene/polymethylmethacrylate (PS/PMMA) (70/30) blend compatibilized with PS-*b*-PMMA block copolymers and by Marić and Macosko [93] in the polystyrene/polydimethylsiloxane (PS/PDMS) (80/20) blends compatibilized with the PS-*b*-PDMS diblock copolymers. PMMA has substantially large viscosity than PS with the same molecular weight under the mixing condition [72]. For the blends with a high viscosity ratio of the dispersed phase to the matrix, *p* = *η*_d_/*η*_m_, the addition of 5% of PS-*b*-PMMA does not reduce the size of the PMMA particles. It is apparently caused by a very low efficiency of the flow-induced coalescence in non-compatibilized blends and the small effect of a compatibilizer on the critical droplet size for breakup, as is discussed in Section 4. What is surprising is a substantial growth of the size of the PMMA droplets after the addition of PS-*b*-PMMA with the block substantially longer than the homopolymers. For the blends PS/PMMA with a comparable viscosity of the components, the addition of the block copolymers reduces the size of the PMMA particles after mixing. On the other hand, only sufficient amounts of the copolymers with the length of the blocks comparable to the homopolymers stabilized the particle size during annealing. Addition of the block copolymers reduced the particle size in the blends of the long PS chains with the short PMMA chains with the viscosity ratio of the components about 0.3. However, none of added copolymers stabilized the size of the PMMA droplet during annealing.

PDMS of the molecular weight comparable to PS had the substantially lower viscosity under the mixing conditions [93]. For blends with a very low viscosity ratio, only PS-*b*-PDMS with both blocks shorter than the related homopolymers reduced substantially the particle size after mixing; the longer copolymers were inefficient. On the other hand, only the block copolymer with the S block somewhat shorter than the PS and DMS block comparable to the PDMS stabilized the size of the PDMS particles during annealing. It is surprising that the block copolymer inhibiting the coalescence during annealing does not affect the droplet size in flow. The same block copolymer reduced the particle size and stabilized it during annealing also in the blends of the same PS and somewhat longer PDMS with the viscosity ratio about 0.3. For the blends of the same PS with the substantially longer PDMS having the viscosity ratio 2.6, all block copolymer reduced the particle size at the end of mixing. However, only the block copolymers of medium molecular weight with the S blocks longer than the PS and DMS blocks substantially shorter than the PDMS stabilized the size of the PDMS particles during annealing. The described effects are simplified and summarized in Table 3.

In interpreting the effect of a copolymer on the size of the dispersed droplets formed in flow, it is impossible to separate the copolymer effect on the droplet breakup and on the flow-induced coalescence for blends with a certain composition. It should be mentioned that the reduction in the droplet size through the copolymer effect on the flow-induced coalescence is achieved when the slow-down of the matrix drainage between the colliding droplets is sufficient for the remarkable decrease in the collision efficiency. On the other hand, the stability of the droplet size during annealing is only achieved when the approach of the droplets covered with a copolymer is negligible during tens of minutes.

### 7.3. Distribution of a Copolymer in Polymer Blends in Steady Flow and Quiescent State

Interesting data were obtained for the PS/polyolefin (PO) blends compatibilized with styrene-butadiene (SB) block copolymers, where the localization of SB copolymers can be easily detected by the transmission electron microscopy [94,95,96,97,98,99,100,101,102,103]. In addition to the transmission electron microscopy, the scanning electron microscopy, providing the blend morphology on a larger scale, and the small angle X-ray scattering, detecting the internal ordered structure of the SB copolymers, were applied. The diblock (S-B), triblock (S-B-S) and pentablock (S-B-S-B-S) laboratory-prepared copolymers with the theoretical relative molecular weight of the butadiene block 60000 and of the styrene block 10000 or 40000 were used in most studies. In some papers, the compatibilization efficiency of other commercial and laboratory prepared SB copolymers were studied. The length of the polyolefin chains was always longer than that of the butadiene block, the polystyrene chains were longer than both the short and long styrene blocks. The measurements of rheological properties of the used SB copolymers showed that most of them were below the order–disorder transition temperature under the mixing condition [97]. However, they frequently served as quite efficient compatibilizers. This demonstrates only limited validity of Chun’s and Han’s rule [14] that only the copolymers above the order–disorder transition temperature can serve as compatibilizers.

A part of the observed differences between the localization of the SB copolymers with the short and long styrene blocks in the PS/polypropylene (PP) [99,103], PS/high-density polyethylene (HDPE) [95] and PS/low-density polyethylene (LDPE) [97] (4/1) blends can be explained reasonably using the assumption that the distribution of the SB copolymer between the bulk phases and the interface is controlled by the rules of equilibrium thermodynamics. Besides the SB envelopes (various thickness and continuity), the SB copolymers with short S blocks form submicron particles mostly localized at the interface. On the other hand, SB copolymers with long S blocks are localized in the PS matrix in the form of very small particles (see Figure 8).

The scattering measurements showed that a part of the SB copolymers with short S blocks kept their internal structure in contrast to those with long S blocks. SB copolymers with short S blocks are apparently poorly soluble in the polyolefin phase due to positive *χ* parameter between PO and PB and in the PS phase due to low content of PS in SB. Therefore, a part of SB is dispersed into small particles keeping the structure of neat SB. The solubility of SB with long S blocks in PS is much better than those with short S blocks. Therefore, these SB copolymers are dissolved in the PS phase and start to form micelles above the critical micellar concentration. The S-B-S triblock copolymer with long styrene blocks reduced the size of the polybutadiene particles in the PS/PB (4/1) blend much more efficiently than that with short styrene blocks [102]. A part of SB with long S blocks was localized as small particles, not showing an internal structure of the neat copolymer, in PS. Due to the impossibility of discriminating between the PB and SB phases, localization of SB with short S blocks in the PS/PB blend was not detected. However, the X-ray scattering showed that a part of this SB kept its own internal structure in the blend.

The interpretation of the effect of SB copolymers on the morphology of the PS/PO (1/4) blends is somewhat more complicated. The localization and structure of the SB copolymers in the PS/PP (1/1) and (1/4) blends did not differ qualitatively from those in the PS/PP (4/1) blend [99]. In contrast to the PS/LDPE (4/1) blends, the compatibilized PS/LDPE (1/4) blends did not show the SB envelopes around the dispersed PS particles [96]. The SB copolymers with short styrene blocks were dispersed in the LDPE matrix as submicron particles. The SB diblock and triblock copolymers with long S blocks were localized in the form of very small particles in the PS dispersed phase and as somewhat larger (yet submicron) particles in the LDPE matrix. The pentablock SB with long S block was localized in the form of quite large (micron-size) pieces in the dispersed PS particles. The internal structure of the related SB copolymers was detected in the blends compatibilized with the triblock and pentablock copolymers with short and long S blocks, but not in the blends compatibilized with the diblock copolymers. It seems that the differences between the localization of the SB copolymers in the PS/LDPE (4/1) and (1/4) blends cannot be explained solely by the equilibrium thermodynamics.

A remarkable decrease in the size of the LDPE droplets was detected when the concentration of SB copolymers with the long S blocks in the compatibilized PS/LDPE (4/1) blends increased from 25 to 50% per the LDPE dispersed phase [97] in spite of the fact that a substantial part of the SB copolymer was localized in the bulk PS phase already with the addition of 25% of SB per LDPE. The same decrease in the size of the PP droplets was found for the PS/PP (4/1) blends compatibilized with the triblock and heptablock SB copolymer [103]. Note that 25% is substantially more than the maximum *C*_crit_ observed by Favis’ group [89,91]. An unexpected difference between the morphology of LDPE/PS (80/20) [96] and LDPE2/PS (75/25) [101], where the viscosity of LDPE was higher than that of LDPE2 compatibilized with SB, was observed. Besides the pentablock with long S blocks, all SB copolymers reduced the size of the PS particles substantially more efficiently in the LDPE2/PS (75/25) blends than in LDPE/PS (80/20) blends.

The comparison of the morphology of the quenched, compression-molded and annealed samples of the PS/PO/SB blends showed an intensive development of the molten blend morphology in the quiescent state [94,97,98,101]. The PS particles in the compression-molded samples of the LDPE2/PS (75/25) blends compatibilized with the diblocks with short and long S blocks was split by the SB partitions [101]. The comparison with the samples quenched immediately after mixing, showing small PS particles, indicating that the PS particles split by the partitions were formed by the coalescence of small PS particles surrounded by the SB copolymers during the compression molding (see Figure 9).

A formation of honeycomb-like particles split the SB copolymers during the compression molding and their further growth during the annealing from small particles prepared by the melt mixing was observed for the PP/PS (4/1) blends compatibilized with various SB copolymers [98]. Growth of the PS particles during the compression molding and the further annealing was detected also for the PP/PS/SB blends where honeycomb-like structure did not form [98]. These results support the conclusion of Macosko’s group that the copolymers efficiently reducing particle size during mixing need not suppress coalescence in quiescent state [71,72,93]. Note that dramatic differences between the morphology of quenched and compression-molded samples of the compatibilized blends were observed.

A co-continuous structure containing the SB particles in the LDPE phase was formed during the mixing in the PS/LDPE (4/1) blend compatibilized by the S-B-S copolymer with short S blocks [97]. A continuous structure of LDPE burst during compression molding and LDPE particles surrounded by the continuous SB envelop with some small SB particles were formed. During the annealing, the size of the LDPE particles increased and some SB particles ended up in the LDPE phase. Small PS particles, partly surrounded by SB, were found in the quenched samples of the PS/LDPE (4/1) blend compatibilized with S-B-S that contains long S blocks. The size of the LDPE particles increased and the SB envelope of the particles developed during the compression molding. The other part of SB was dispersed in the PS matrix. The further increase in the size of PS particles was observed after the annealing. A migration of the SB triblock copolymer with a long S block from the PS matrix to the interface was observed during annealing of the compatibilized PS/PP (4/1) blends [99].

In the PS/HDPE (4/1) blends compatibilized with the S-B diblock containing a short S block, the SB copolymer formed a continuous envelope around the HDPE particles in the samples mixed at 60 rpm but it was located at the interface in the form of small droplets in the samples mixed at 120 rpm. The dependence of the localization of the SB copolymers in the PO/PS blends on the rate of mixing together with the detected migration of SB copolymer between the interface and bulk phases during the annealing show that the compatibilization efficiency of a copolymer calculated using the equilibrium thermodynamics can serve as a rough estimation only. The morphology evolution in the molten quiescent compatibilized blends needs further theoretical and experimental studies.

## 8. Summary of the Present State of the Art

The confrontation of the phenomenological predictive rules for the compatibilization efficiency of the copolymers with experimental data shows that the validity of these rules is limited.

The comparison of experimental data with the available theories shows that the minimum Gibbs energy rule does not provide the rigorous distribution of a copolymer between the interface and bulk phases in flowing immiscible polymer blends. This rule, however, can serve for the semiquantitative prediction of the distribution.

The distribution of a copolymer between the interface and bulk phases has to be determined before the calculation of the effect of an added copolymer on the interfacial tension in immiscible blends. Neither the assumption that the amount of a copolymer dissolved in the bulk phases is equal to the amount of an added copolymer nor the assumption that the whole amount of an added copolymer is localized at the interface is applicable to most polymer blends.

Recent calculations showed that a low concentration of a copolymer at the interface, where the copolymer blocks were not stretched, already reduced the interfacial tension to zero in many systems. The wet-mushroom model (a low amount of a copolymer at the interface), not the dry- or wet-brush model, is adequate for these systems.

Experimental results support the presumption that a low solubility of a copolymer in the bulk phases causes the existence of small pieces of a copolymer keeping its internal structure in compatibilized blends. A large concentration of a copolymer that is well-soluble at least in one bulk phase leads to the formation of micelles in the bulk phases.

A limited knowledge of the diffusion of a copolymer along the interface and between the interface and bulk phases is an obstacle to derivation of a reliable theory of the droplet breakup in compatibilized polymer blends. Experimental data indicate that the effect of the decreasing interfacial tension on a droplet radius is almost compensated by the effect of the increasing critical capillary number for many compatibilized polymer blends.

Two negatively correlated mechanisms suppress the efficiency of the coalescence in compatibilized polymer blends with respect to non-compatibilized blends. The Marangoni stress slows down the matrix drainage between colliding droplets due to a decrease in the mobility of the interface. Its value increases with the rate of droplets approach, i.e., the shear rate for the flow-induced coalescence. The reliability of the calculation of the Marangoni effect is affected by limited knowledge of a copolymer diffusion along the interface and between the interface and bulk phases. The steric repulsion manifests in the inter-droplet distances comparable to the copolymer radius of gyration; it opposes the approach of droplets to the distance necessary for the matrix film rupture. It does not depend on the rate of droplets approach.

The difference between the efficiency of various copolymers in suppression of the flow-induced coalescence and the coalescence in a quiescent state is caused by the differences in forces causing the droplet approach in these cases. The efficiency of the flow-induced coalescence is controlled by the competition between the droplet approach and rotation. Therefore, a decrease in the rate of the droplets approach caused by the presence of a compatibilizer together with its negligible effect on the droplets rotation can lead to a dramatic decrease in the coalescence efficiency. On the other hand, the attractive molecular forces and the Brownian motion induce the coalescence in the quiescent state. In this case, a decrease in the rate of droplets approach due to the effect of a copolymer on the matrix drainage manifests by a continuous slowdown of the rate of coalescence. Note that the knowledge of effective molecular forces between droplets enveloped by a copolymer is necessary to derive the theory of coalescence in quiescent compatibilized polymer blends.

The experiments detected formation of relatively large partitioned particles during annealing from small droplets surrounded by a copolymer. The particles formation can be explained by the above-mentioned differences between the mechanisms of the coalescence in flow and in the quiescent state. The submicron droplets, formed during mixing due to efficient suppression of the flow-induced coalescence, attract each other quite rapidly due to their small size. The steric repulsion of the touching droplets hinders the rupture of the interfacial layer between them. On the other hand, no force inducing separation of the partitions exists in quiescent polymer blends.

## 9. Conclusions

A description of the effect of a compatibilizer on the phase structure formation in flowing immiscible polymer blends needs the solution of several non-trivial tasks: distribution of a copolymer between the interface and bulk phases, the effect of a copolymer on the interfacial tension, the effect of a copolymer on the droplet breakup and coalescence and their competition in flow. The solutions of equations describing these individual effects are inter-dependent. Therefore, a general theory of the phase structure evolution in compatibilized polymer blends has not been formulated so far. On the other hand, approximate theories of individual events contribute to a more reliable interpretation of experimental results.

Further development of theories of individual steps of the phase structure evolution in compatibilized polymer blends with careful comparison of these results with experimental data is needed for reliable prediction of the dependence of a copolymer compatibilization efficiency on its architecture.

## Figures and Tables

**Figure 1 materials-14-07786-f001:**
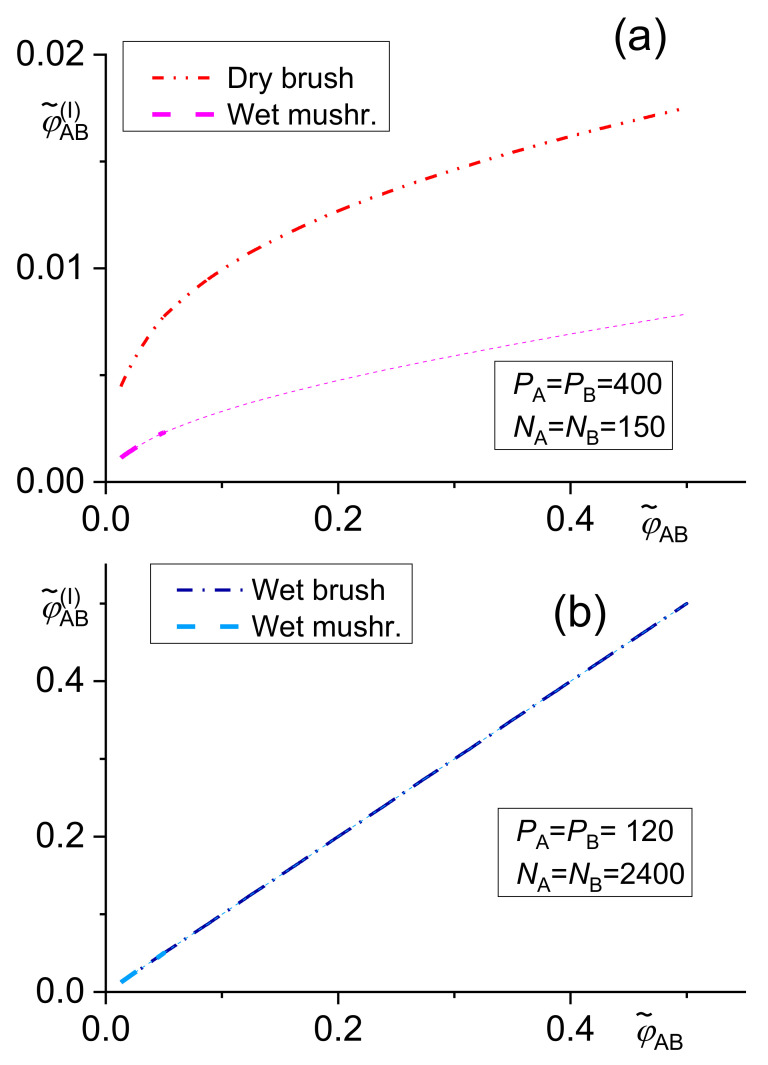
Dependence of the copolymer content at the interface ${\tilde{\varphi }}_{\mathrm{CD}}^{\left(I\right)}$ on the total copolymer content ${\tilde{\varphi }}_{\mathrm{CD}}$ calculated using Equations (7)–(9): (**a**) for the total copolymer block lengths *N*_C_ = *N*_D_ = 150, homopolymer chain lengths *P*_A_ = *P*_B_ = 400 (the chain length ratio typical of the dry brush regime), (**b**) for the total copolymer block lengths *N*_C_ = *N*_D_ = 2400, homopolymer chain lengths *P*_A_ = *P*_B_ = 120 (the ratio of the wet brush regime)., The Flory–Huggins interaction parameter was *χ* = 0.04 Interfacial area to volume parameter *Γ* = 0.00525. Volume fraction of the dispersed phase: *φ*_A_ = 0.3; segment length *a* = 0.7 nm. The thicker part of the wet mushroom curve presents the region where the condition of this model is met; for the dry brush regime, not all conditions are met.

**Figure 2 materials-14-07786-f002:**
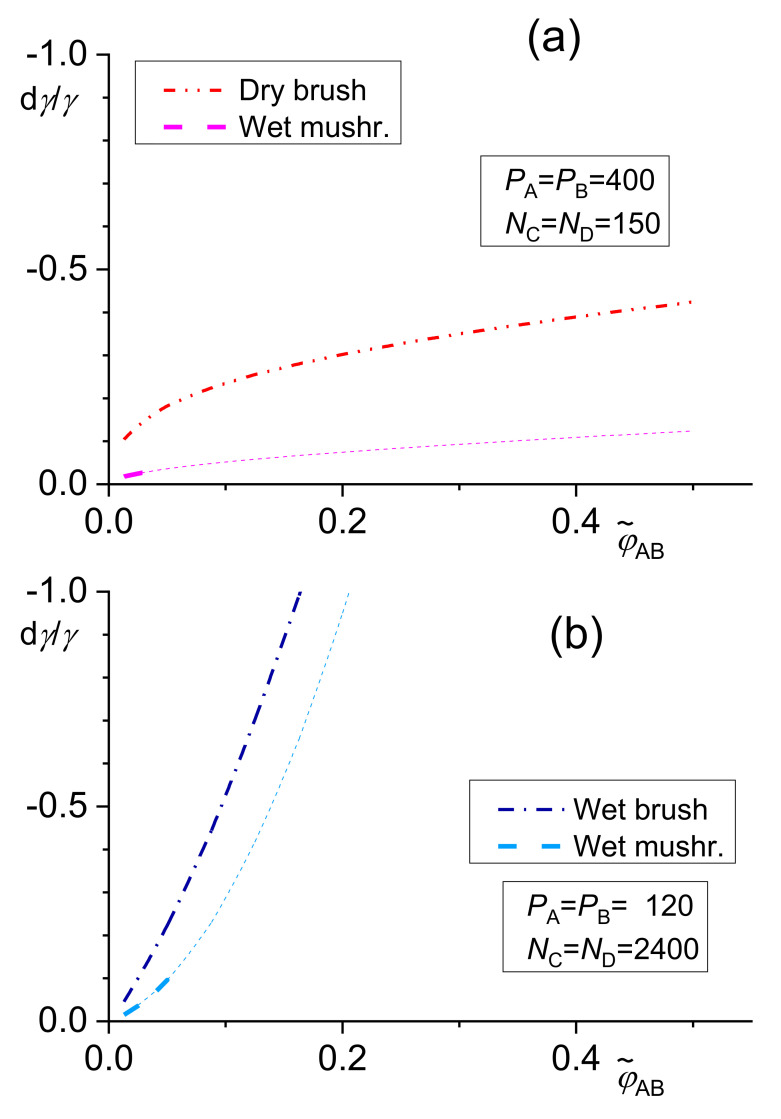
Dependence of the relative decrease in the interfacial tension *γ* on the total copolymer content ${\tilde{\varphi }}_{\mathrm{CD}}$ calculated using Equations (12)–(14). Parameters and the meaning of (a) = homopolymer is longer and (b) = copolymer is longer are the same as in Figure 1.

**Figure 3 materials-14-07786-f003:**
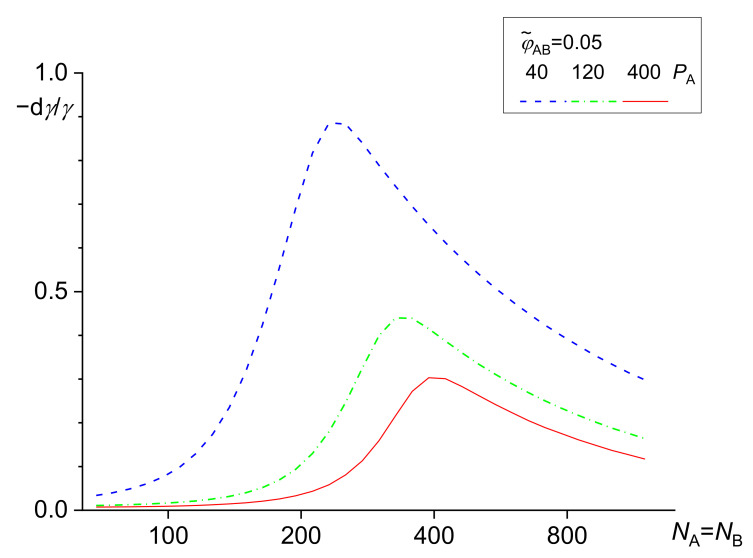
Dependence of the relative decrease in the interfacial tension *γ* on the copolymer blocks length *N*_A_ = *N*_B_ in the range from 60 to 1200 for copolymer volume fraction ${\tilde{\varphi }}_{\mathrm{AB}}^{\left(I\right)}$ = 0.05 and homopolymer lengths *P*_A_ = *P*_B_ of 40, 120 and 400. Other parameters are the same as in Figure 1. Equation (12) for the low copolymer content was used.

**Figure 4 materials-14-07786-f004:**
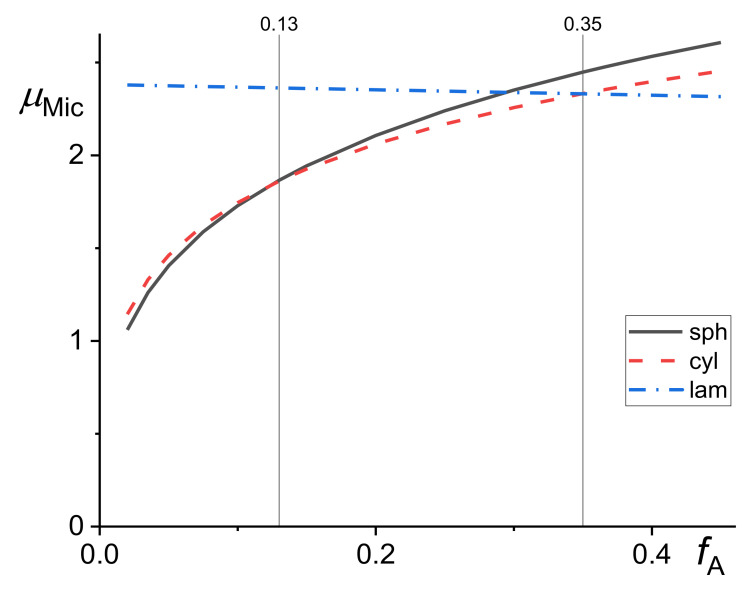
Dependence of the chemical potentials of spherical, $\mu_{\mathrm{Mic}}^{\mathrm{sph}}$, cylindrical, $\mu_{\mathrm{Mic}}^{\mathrm{cyl}}$, and lamellar, $\mu_{\mathrm{Mic}}^{\mathrm{lam}}$, micelles on the length proportion of the A chain in the copolymer *f*_A_ for the total copolymer chain length *N* = 200 and the Flory–Huggins interaction parameter *χ* = 0.04 calculated using Equations (15)–(17).

**Figure 5 materials-14-07786-f005:**
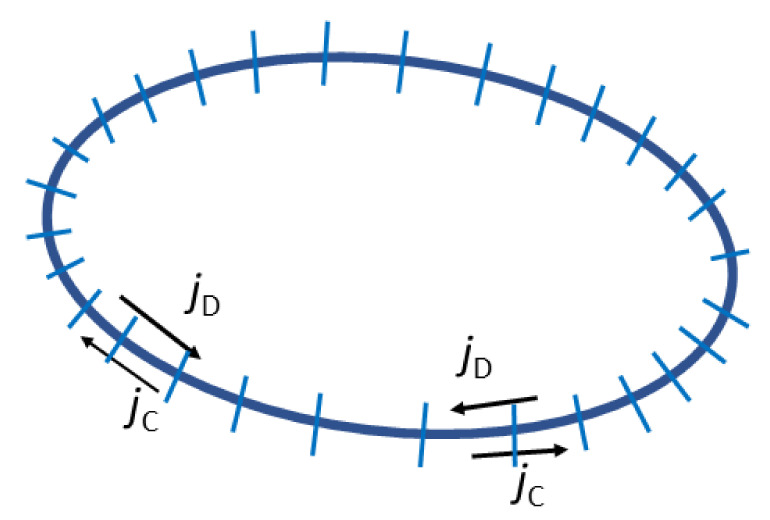
Migration of a copolymer on the droplet surface in shear flow. *j*_C_ is the convective flux, *j*_D_ the diffusive flux.

**Figure 6 materials-14-07786-f006:**
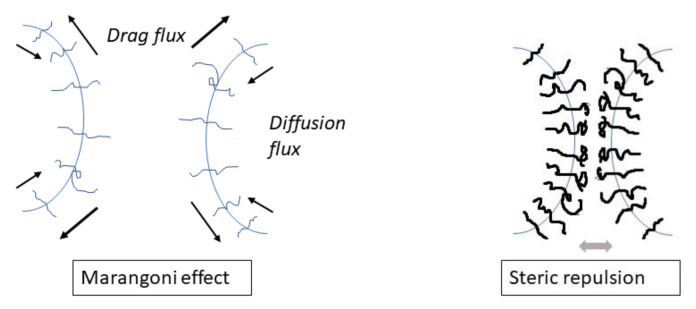
Two main mechanisms of the coalescence suppression.

**Figure 7 materials-14-07786-f007:**
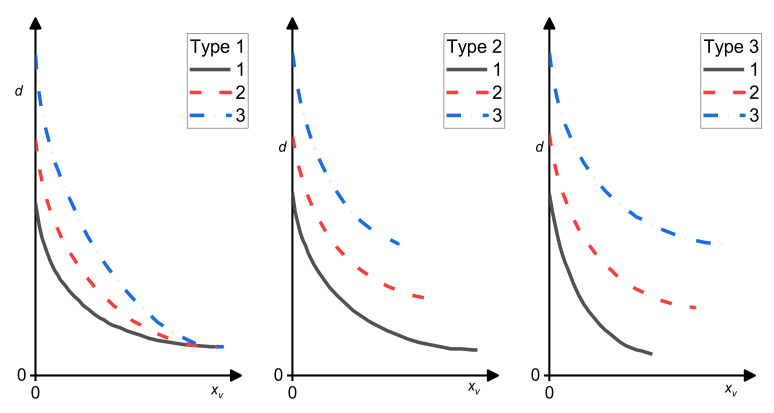
Three types of emulsification curves (dependence of the drop diameter *d* on the copolymer content *x_V_*) redrawn according to Favis [91]. Concentration of the dispersed phase relating to the individual curves increases in the order of their numbering.

**Figure 8 materials-14-07786-f008:**
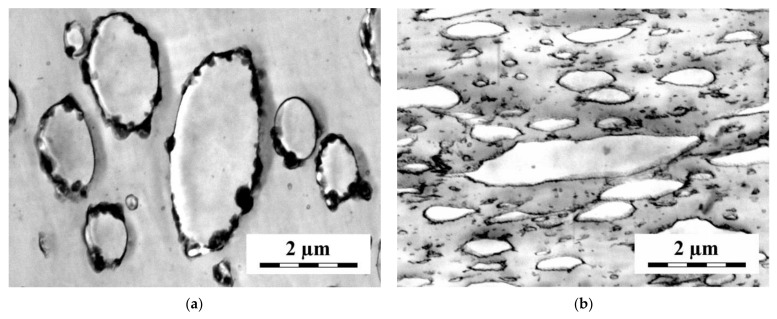
Localization of SB compatibilizer with the short (**a**) and long (**b**) styrene blocks in compression molded PS/LDPE (80/20) blend (up to now unpublished photo of the system studied in ref. [97]).

**Figure 9 materials-14-07786-f009:**
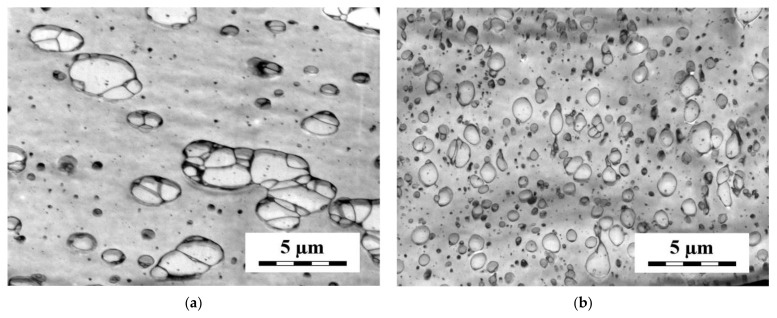
Morphology of compression molded (**a**) and quenched (**b**) samples of LDPE/PS (75/25) blend compatibilized with 5% of SB copolymer with the short styrene block (up to now unpublished photo of the system studied in ref. [101]).

**Table 1 materials-14-07786-t001:** Comparison of the conditions under which individual approximations of the free energy of copolymer brushes are applicable.

Model	Density of a Copolymerat the Interface	Stretching of Copolymer Blocks	*N_i_*/*P_i_* Ratio for Applicability of the Model
Wet mushroom	Very low	Unstretched	Whole range
Wet brush	Low	Stretched	High, *N**_i_* > *P**_i_*^3/2^
Dry brush	High	Stretched	Small, *N**_i_* < *P**_i_*^3/2^

**Table 2 materials-14-07786-t002:** Comparison of the main features of the Marangoni effect and steric repulsion.

Feature	Distance of Initiation	More Important for	Dependence on the Rate of Droplets Approach	Note
Marangoni effect	Droplet radius	Low copolymer density at the interface	Increase with the rate	
Steric repulsion	Copolymer end-to-end distance	High copolymer density at the interface	Independent of the rate	Negative correlation with the Marangoni effect

**Table 3 materials-14-07786-t003:** Simplified comparison of copolymer effect on droplet size for systems of different type. *p* is the droplet to matrix viscosity ratio, *N* the copolymer chain length, *N_i_* and *P_i_* length of the units *i* block in copolymer or chain of homopolymer.

PS Matrix—PMMA Particles		Affecting Particle Size	
*p = η*_d_/*η*_m_	*N_i_* vs. *P_i_*	During Mixing	After Annealing
10 to 30	*N_i_* ≈ *P_i_*	No effect	Almost stable
10 to 30	*N_i_* >> *P_i_*	Increases	Does not stabilize
0.3 to 2		Reduces	Does not stabilize
**PS Matrix—PDMS Particles**		**Affecting Particle Size**	
* **p** *	* **N** _ **i** _ * **vs.** * **P** _ **i** _ *	**During Mixing**	**After Annealing**
0.03 to 0.05	*N_i_* < *P_i_*	Reduces	Does not stabilize
0.03 to 0.05	*N_i_* > *P_i_*	no	Does not stabilize
0.3	*N_i_* ≈ *P_i_*	no	Stabilizes
2.6	*N*_S_ > *P*_PS_, *N*_DMS_ << P_PDMS_	Reduces	Stabilizes
2.6	Other than above	Reduces	Does not stabilize

## Data Availability

No own data are associated with this review.

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
