# Peer review of "The Effects of Copolymer Compatibilizers on the Phase Structure Evolution in Polymer Blends—A Review"

_materials, 2021, doi:10.3390/ma14247786_

Round 1
Reviewer 1 Report
In this paper, the effects of copolymer compatibilizers on the phase structure evolution in polymer blends are reviewed in detail from the perspectives of theory and experiment. According to thermodynamic rules, the minimum Gibbs energy rule just provide the semiquantitative prediction of the distribution of copolymer between the interface and bulk phases in flowing immiscible polymer blends.
Based on this, some other theories are proposed to calculate the distribution of copolymer. The effect of copolymer fraction and the copolymer geometry on the distribution and the interfacial tension have been studied.
Furthermore, how the existence of compatibilizers affects the structure formation of blends is described from two aspects: droplet breakup and coalescence. To sum up, the role of compatibilizer in blends has been discussed relatively completely, and some experimental data can correspond with the theory. However, the experimental part can add more content, such as different processing methods, different chain segment length, the proportion of copolymers (or the chain segment length of the co-polymer unit) , even the crystallization, all of these will affect the formation and stability of the phase structure. So It is recommended that the article will be accepted after minor revision.
Author Response
Dear Reviewer,
a note briefly discussing plausibility of relations between theory and experimental data has been added to the first paragraph of the Section 7. Please, if it does not meet your expectations, we welcome more specific comments.
Sincerely,
Josef Jůza
Reviewer 2 Report
This manuscript reviewed the effect of block and graft copolymers on the phase structure in immiscible polymer blends from theoretical and experimental consideration. I recommend to publish it after revision.
1) In Equation 1 , some parameters should be defined.
2) Please define the terms in this work, like order-disorder transition temperature.
3) Provide the physical interpretation for some symbols in Figures for more accurate and clear understanding.
4)The most important thing is that the polymer blends can be classified into amorphous/amorphous blends, amorphous/crystalline blend and crystalline/crystalline blends. The phase structure of the blends at melt states should be discussed which might exclude the influence of crystallization on phase structures.
5) As the authors discussed in the manuscript “this paper also deals with the effect of a copolymer on the phase structure evolution in the blends with the droplets in the matrix morphology “
6) The authors are suggested to simplify the conclusions and point out clearly the effects of copolymers on the blends.
Author Response
Dear Reviewer,
thank you for your suggestions helping to improve the article. Answers to your comments follow:
1) In Equation 1, some parameters should be defined.
Answer: Parameters γ0 (interfacial tension of non-compatibilized blend) and e (the Euler number) have been defined.
2) Please define the terms in this work, like order-disorder transition temperature.
Answer: Terms order-disorder transition temperature, Marangoni effect and steric repulsion have been briefly explained in the revised manuscript.
3) Provide the physical interpretation for some symbols in Figures for more accurate and clear understanding.
Answer: In Figures 1 and 4, both symbols and quantities names are now in the legend. The Γ parameter in now accompanied by a short description, while we think that the Flory-Huggins interaction parameter need not be explained.
4)The most important thing is that the polymer blends can be classified into amorphous/amorphous blends, amorphous/crystalline blend and crystalline/crystalline blends. The phase structure of the blends at melt states should be discussed which might exclude the influence of crystallization on phase structures.
Answer: A text explaining that the paper is focused on the effect of copolymers in the melt (where the structure of blends in the solid state is irrelevant) has been added to the Introduction.
5) As the authors discussed in the manuscript “this paper also deals with the effect of a copolymer on the phase structure evolution in the blends with the droplets in the matrix morphology “
Answer: The sentence has been corrected to "this paper mostly deals with..." in the revised manuscript. The reasons, why the effect of a copolymer on the other types of the blend phase structure is not discussed, are explained in the Introduction.
Since most changes are results of the suggestions of more reviewers, they are not specifically distinguished according to reviewer.
Sincerely,
Josef Jůza
Reviewer 3 Report
Dear Editor and Authors,
I read carefully the review entitled ‘’ The Effects of Copolymer Compatibilizers on the Phase Structure Evolution in Polymer Blends’’. I can conclude that is an interesting paper, but some issues must be clarified:
- In Figure 3, the steric repulsion must be presented more evidently.
- The English must be polished.

Author Response
Dear Reviewer,
thank you for your suggestions. Our answers are:
In Figure 3, the steric repulsion must be presented more evidently.
Answer: The picture of the steric repulsion has been corrected, a steric repulsion has been briefly defined in the revised text.
The English must be polished.
Answer: The English has been again checked, several dozens of tiny corrections are marked in the revised manuscript.
Sincerely,
Josef Jůza
Reviewer 4 Report
This work provides a detailed theoretical and experimental description of the evolution of the phase structure in immiscible binary polymer blends in the presence of a common compatibilizing agent consisting of a block copolymer.
Specifically, the authors carefully reviewed the state of the art of the knowledge already available on this issue highlighting, among other things, the strengths and limitations of the theories already developed and comparing theoretical results with properly selected experimental data.
Despite the complexity of the topics covered, the text appears fluid and easy to understand even for readers who are less familiar with the aspects covered.
Recognizing the scientific relevance of this work, which is capable of allowing a rapid transfer of knowledge on a complex topic, which has been widely addressed in the last thirty years, its publication is firmly recommended.
Author Response
Dear Reviewer,
we thank to the reviewer for the positive opinion; I have found nothing requiring the manuscript modification in it.
Sincerely,
Josef Jůza
Reviewer 5 Report
COMMENTS FOR AUTHORS:
- Bibliography is not taken from recent work.
- Many equations are used but their physical significance must be illustrated using necessary figures wherever required.
- All the equations taken from the reported literature must be cited properly.
- Mechanisms of the copolymer effect should in tabulated form along with their benefits and problems.
- The present “Conclusions” section seems summary of the review. So, it may be renamed as “Summary”.
- The present “Conclusions” is too large. The “Conclusions” should be more focused and precised.
Author Response
Reviewer 5:
Dear Reviewer,
thank you for your comments. They were less pleasant, but most beneficial for the article.
- Bibliography is not taken from recent work.
Answer: Several new references have been added and their contents have been shortly reported in the article, where applicable.
- Many equations are used but their physical significance must be illustrated using necessary figures wherever required.
Answer: Three figures (1, 2 and 4 in the new version) have been added.
- All the equations taken from the reported literature must be cited properly.
Answer: References to the sources have been added before each equation.
- Mechanisms of the copolymer effect should in tabulated form along with their benefits and problems.
Answer: Three comparing tables (1-3) have been added.
- The present “Conclusions” section seems summary of the review. So, it may be renamed as “Summary”.
Answer: Renamed according to the suggestion.
- The present “Conclusions” is too large. The “Conclusions” should be more focused and precised.
Answer: New Conclusions section has been added.
We are ready to make additional changes according to further specifications.
Sincerely,
Josef Jůza
Round 2
Reviewer 5 Report
The revised manuscript has been corrected significantly. Thus, it may be accepted for publication.